# Is *Hyperdermium* Congeneric with *Ascopolyporus*? Phylogenetic Relationships of *Ascopolyporus* spp. (*Cordycipitaceae*, *Hypocreales*) and a New Genus *Neohyperdermium* on Scale Insects in Thailand

**DOI:** 10.3390/jof8050516

**Published:** 2022-05-17

**Authors:** Donnaya Thanakitpipattana, Suchada Mongkolsamrit, Artit Khonsanit, Winanda Himaman, Janet Jennifer Luangsa-ard, Natapol Pornputtapong

**Affiliations:** 1Department of Biochemistry and Microbiology, Center of Excellence for DNA Barcoding of Thai Medicinal Plants, Faculty of Pharmaceutical Sciences, Chulalongkorn University, Bangkok 10330, Thailand; 6176134833@student.chula.ac.th or; 2Plant Microbe Interaction Research Team, National Center for Genetic Engineering and Biotechnology (BIOTEC), 113 Thailand Science Park, Phahonyothin Road, Khlong Nueng, Khlong Luang, Pathum Thani 12120, Thailand; suchada@biotec.or.th (S.M.); artit.kho@biotec.or.th (A.K.); 3Forest Entomology and Microbiology Research Group, Forest and Plant Conservation Research Office, 61 Department of National Parks, Wildlife and Plant Conservation, Phahonyothin Road, Chatuchak, Bangkok 10900, Thailand; winandah@gmail.com

**Keywords:** *Cordycipitaceae*, epiphyte, insect pathogenic fungi, multigene phylogeny, scale insects

## Abstract

During surveys of insect pathogenic fungi (IPF) in Thailand, fungi associated with scale insects and plants were found to represent five new species of the genus *Ascopolyporus* in *Cordycipitaceae*. Their macroscopic features resembled both *Hyperdermium* and *Ascopolyporus*. Morphological comparisons with the type and known *Ascopolyporus* and *Hyperdermium* species and phylogenetic evidence from a multigene dataset support the appointment of a new species of *Ascopolyporus*. Moreover, the data also revealed that the type species of *Hyperdermium*, *H. caulium*, is nested within *Ascopolyporus*, suggesting that *Hyperdermium* is congeneric with *Ascopolyporus*. The specimens investigated here differ from other *Ascopolyporus* species by phenotypic characters including size and color of stromata. Phylogenetic analyses of combined LSU, *TEF1*, *RPB1* and *RPB2* sequences strongly support the notion that these strains are distinct from known species of *Ascopolyporus*, and are proposed as *Ascopolyporus albus*, *A. galloides*, *A. griseoperitheciatus*, *A. khaoyaiensis* and *A. purpuratus*. *Neohyperdermium* gen. nov. is introduced for other species originally assigned to *Hyperdermium* and *Cordyceps* occurring on scale insects and host plants as epiphytes, accommodating two new combinations of *Hyperdermium pulvinatum* and *Cordyceps piperis*.

## 1. Introduction

Scale insects are a diverse group of sap-sucking insects in the superfamily *Coccoidea* of the order *Hemiptera*, associated with aphids (*Aphidoidea*) and whiteflies (*Aleyrodoidea*) [1,2]. These insects cause damage by sucking fluids from leaves, stems and other parts of host plants and excrete honeydew that favors sooty mold growth, which consequently decreases photosynthetic rates. They belong to seven families: *Antennulariellaceae*, *Capnodiaceae*, *Chaetothyriaceae*, *Coccodiniaceae*, *Euantennariaceae*, *Metacapnodiaceae* and *Trichomeriaceae* [3,4,5]. In addition, many groups of fungi are known to grow on various scale insects by covering the whole surface of the insect body and can be found in the phyla (a) *Basidiomycota*: *Septobasidiales* (*Septobasidium* and *Uredinella*), (b) *Chytridiomycota*: *Blastocladiales* (*Myiophagus*) and (c) *Ascomycota*: *Myriangiales* (*Myriangium*), *Pleosporales* (*Podonectria*), and especially in a large group of entomopathogens in the *Hypocreales* [6,7,8,9].

Hypocrealean fungi associated with armored (*Diaspididae*) and soft-scale insects (*Coccidae*) can be found in various genera within five families: (1) *Bionectriaceae* viz. *Clonostachys* Corda; (2) *Nectriaceae* viz. *Microcera* Desm. and *Fusarium* Link; (3) *Cordycipitaceae* viz. *Ascopolyporus* Möller, *Cordyceps* Fr. and *Hyperdermium* J.F. White, R.F. Sullivan, Bills and Hywel-Jones; (4) *Ophiocordycipitaceae* viz. *Ophiocordyceps* Petch; and (5) *Clavicipitaceae* viz. *Aschersonia* Mont., *Conoideocrella* D. Johnson, G.H. Sung, Hywel-Jones and Spatafora, *Dussiella* Pat., *Helicocollum* Luangsa-ard, Mongkols., Noisrip. and Thanakitp., *Hypocrella* Sacc., *Regiocrella* P. Chaverri and K.T. Hodge, *Orbiocrella* D. Johnson, G.H. Sung, Hywel-Jones and Spatafora and *Samuelsia* P. Chaverri and K.T. Hodge [9,10,11,12,13,14,15,16,17,18,19,20,21]. Among them, the most abundant and widespread members are found in *Clavicipitaceae* and *Cordycipitaceae*. The macromorphological characters of these genera in nature can be easily distinguished in each family. Scale insect pathogenic genera in *Clavicipitaceae,* such as *Conoideocrella*, *Hypocrella, Moelleriella* and *Orbiocrella*, possess diverse morphological characters, such as the formation of hard stromata, pulvinate, subglobose or hemispherical and ring-like stromata, as well as the presence of only superficial, cone-shaped perithecia, while in *Cordycipitaceae*, most have pulvinate, subglobose, hemispherical, soft stromata with crowded perithecia. Two different colors are found the upper and lower surface of stromata in some species of *Ascopolyporus* and *Hyperdermium* [11,12,14,16].

*Ascopolyporus* is an epiphytic fungal genus in *Cordycipitaceae* that produces stromata on the stems of living plants as biotrophs and infects scale insects as necrotrophs comprising only seven species [22]. *Ascopolyporus* species are commonly found in tropical forests where bamboo is present [23]. The type species of *Ascopolyporus*, *A. polychrous*, is a pathogen of bamboo scale insects that produces up to 4 cm large subglobose to polypore-like, bright rusty-red or white to yellow perithecial stromata, which are usually fertile only on the underside of the stroma [12,13,24]. In 2005, a new species of *Ascopolyporus*, *A. philodendrus*, was described by Bischoff et al. [14] on bamboo scale insects, and a new description for *A. villosus* was made. They considered that the morphology of perithecial stromata and the conidial states of *Ascopolyporus* resemble the scale insect pathogenic genus *Hyperdermium*, especially its type species, *H. caulium* [11,14]. Both of these species in the two genera share similar morphological characters, having large stromata, immersed perithecia, filiform ascospores and phialidic conidiogenous cells. The anamorph state is referred to as cylindrocarpon-like phialides, characterized by producing multiseptate conidia, a unique character in the *Cordycipitaceae*. Moreover, a species of *Cordyceps*, *C. piperis*, is also capable of parasitizing scale insects but differ by producing verticillium-like anamorph with aseptate conidia [11,12].

During our continuous survey of insect pathogenic fungi (IPF) in national parks and community forests in Thailand, we encountered hyperdermium-like specimens with differences in phenotypic characters including colors and sizes of stromata. These morphologically diverse specimens were preliminarily identified as members of the genus *Hyperdermium* and *Ascopolyporus*. The aims of this study are thus (1) to determine the phylogenetic relationship of these two genera and (2) to identify and describe new species of hyperdermium-like fungi on scale insects from Thailand by combining morphological characteristics and reconstructing their phylogeny based on sequence data of LSU, *TEF1*, *RPB1* and *RPB2* loci.

## 2. Materials and Methods

### 2.1. Collection and Isolation

The 63 epiphytic isolates in this study were collected from various localities in Thailand since June 1992, representing the first recorded collection from Khao Yai National Park, Nakhon Ratchasima Province. Thereafter, these specimens have been found throughout every region in Thailand, albeit not frequently, including the Ban Hua Thung community forest in Chiang Mai Province; Chiang Dao, Khao Soi Dao and Khlong Nakha wildlife sanctuaries; Kaeng Krachan and Khlong Lan national parks; and the Khao Chong wildlife development and conservation promotion station. Specimens were examined for fungal colonization from the stems and leaves of monocotyledonous and dicotyledonous plants. The specimens were collected and stored in plastic boxes before returning to the laboratory for isolation. Pure cultures were made from the isolation of the sexual morph following Luangsa-ard et al. [25]. The cultures and the voucher specimens were deposited in Thailand Bioresource Research Center (TBRC) and BIOTEC Bangkok Herbarium (BBH), Thailand, respectively.

### 2.2. Morphological Study

For obtaining morphological descriptions, all isolates were cultured on oatmeal agar (OA: oatmeal 60 g, agar 12.5 g, in 1 L distilled water, Difco) and potato dextrose agar (PDA: potato 200 g, dextrose 20 g, agar 15 g, in 1 L distilled water) for 14–20 days. Colony morphology was examined for color, size, shape and appearance. Fungal structures of teleomorph and anamorph states were mounted in lactophenol cotton blue solution, and their characters were investigated by light microscopy, as described by Mongkolsamrit et al. [26] and Khonsanit et al. [27]. Sections of the stroma on stems were prepared by using a freezing microtome (Slee Cryostat MEV, Mainz, Germany), and mounted in distilled water and in lactophenol cotton blue solution [28]. The Sixth Royal Horticultural Society (RHS) color chart was used to characterize the colors of fresh specimens and cultures [29]. Twenty to fifty individual length and width measurements were taken, and the amount of variability is provided as average ± standard deviation with absolute minima and maxima in parentheses.

### 2.3. DNA Extraction, PCR and Sequencing

The mycelial mass of fungi was obtained from cultures grown on PDA for 7 days at 25 °C. A modified CTAB protocol used for DNA extraction using polyvinylpyrrolidone instead of β-mercaptoethanol in CTAB buffer and increasing temperature in the incubation process from 60 °C to 65 °C was previously described by Thanakitpipattana et al. [30]. PCR was used to amplify the nuclear ribosomal large subunits (LSU), the region of the elongation factor 1-α (*TEF1*), and the largest and second-largest subunits of RNA polymerase II (*RPB1* and *RPB2*). The reaction mix was prepared in 25 μL volumes containing 1× Dream Taq Buffer (with included 20 mM MgCl_2_), 0.4 M betaine, 200 μM dNTP mix, 0.5 μM of each primer, 1 Unit Dream Taq DNA polymerase (Thermo Scientific, Waltham, MA, USA), 50 ng of DNA template and Milli-Q water. PCR amplifications of four loci were carried out with the following primers: LROR and LR5 for LSU [31,32], 983F and 2218R for *TEF1* [33], CRPB1 and RPB1-Cr for *RPB1* [34], and RPB2-5F2 and RPB2-7Cr for RPB2 [35,36]. The PCR conditions were performed as follows: 94 °C for 3 min, followed by 35 cycles of denaturation at 94 °C for 1 min, annealing at a suitable temperature for 1 min, extension at 72 °C for 1 min and a final extension of 72 °C for 10 min. The annealing temperature of each gene was 50 °C for *RPB1* and *RPB2*, and 55 °C for *TEF1* and LSU. PCR products were purified and subsequently sequenced with PCR amplification primers.

### 2.4. Sequence Alignment and Phylogenetic Analyses

The newly generated sequences from the twelve strains in this study were assembled using BioEdit v. 7.2.5 [37] and then deposited in the GenBank database under the accession numbers of *TEF1* (OL322029-OL322040), LSU (OL322041-OL322052), *RPB1* (OL322053-OL322059) and *RPB2* (OL322060-OL322070) (Table 1). Sequences of each locus were aligned using MUSCLE 3.6 [38] together with other sequences of related taxa from previous studies for phylogenetic analyses (see Table 1), and manually refined to minimize gaps. The concatenated (LSU + *TEF1* + *RPB1* + *RPB2*) sequences were analyzed by maximum likelihood (ML) and Bayesian inference (BI), both on the CIPRES Science Gateway portal [39]. Maximum likelihood analysis was performed with RAxML-HPC2 on XSEDE v.8.2.12 with default parameters [40] using the GTRCAT substitution model with 1000 rapid bootstrap replicates. The program MrModeltest v.2.2 [41] was used to determine the model of evolution under the Akaike Information Criterion (AIC) implemented in PAUP v.4.0a169 [42], which selected SYM + G as the best nucleotide substitution model. The BI analysis was performed using MrBayes on XSEDE v.3.2.7a with default parameters [43]. The Markov Chain Monte Carlo (MCMC) searches were run for 5,000,000 generations with sampling every 1000 generations and a burn-in value of 10%. Nodes were considered as strongly supported with bootstrap and posterior probability values greater than 70% and 0.7, respectively.

## 3. Results

### 3.1. Molecular Phylogeny

The combined four-gene dataset of 54 taxa consisted of 3404 bp (LSU 861 bp, *TEF1* 954 bp, *RPB1* 730 bp, *RPB2* 859 bp). *Flavocillium bifurcatum* and *Flavocillium* sp. in *Cordycipitaceae* were used as an outgroups. Phylogenetic tree topology obtained from ML was similar to the BI analysis. Therefore, only the ML tree is shown (Figure 1). Multigene phylogenetic analyses revealed that the sequenced strains comprise five novel species and are nested with the type and other species of *Ascopolyporus*, *A. polychrous* and *A. villosus*, as well as type species of *Hyperdermium*, *H. caulium*, within the *Ascopolyporus* clade, with strong support (81% ML bootstrap (MLBS) and 0.99 BI posterior probability (BIPP)), as shown in Figure 1. The type species *H. caulium* is clustered within this clade, suggesting that *Hyperdermium* is congeneric with *Ascopolyporus*, although with low internal bootstrap support because only LSU sequence data are available (<50 MLBS and <0.5 BIPP, data not shown).

Three of our new species are found in the pulvinate subclade showing irregularly subglobose to globose stromata, namely, *Ascopolyporus albus*, *A. galloides* and *A. griseoperitheciatus*, with 93% MLBS and 0.97 BIPP. Another subclade comprises both flattened and pulvinate stromata of two new and known species, including *Ascopolyporus khaoyaiensis*, *A. purpuratus*, *A. polychrous*, *A. villosus* and *H. caulium* (Figure 1). The *Ascopolyporus* clade is sister to the *Blackwellomyces* clade, which produces similar types of phialides and conidial arrangement as well as acremonium-like or lecanicillium-like anamorphs.

The position of *Hyperdermium pulvinatum* and *Cordyceps piperis*, on the other hand, is clearly distant from the *Ascopolyporus* clade, and these two species always clustered together separate from the type species of *Hyperdermium*, *H. caulium*. These two species form a basal clade to *Akanthomyces*, *Samsoniella*, *Beauveria* and *Cordyceps*. Based on their multigene phylogenetic position presented in this study, we propose to transfer these two species to the genus *Neohyperdermium*.

### 3.2. Taxonomy

***Ascopolyporus*** Möller **emend**. Thanakitpipattana and Luangsa-ard.

Stromatal mass exceeding scale insect host. *Stroma* bulbous (lumpy or tuberous) or ungulate, flattened or pulvinate, fleshy or gall-like, polypore-like, white, yellowish white, purple to orange; sterile surface and fertile underneath the stroma. *Perithecia* semi-immersed to immersed, ovate to obclavate or cone-shaped. *Asci* hyaline, filiform. *Ascospores* hyaline, whole with septation or aseptate. Conidiogenous cells phialidic, solitary, slightly curved. *Conidia* hyaline, fusiform to subcylindrical, acerose, aseptate or 1–5 septate when mature, in chains or in sticky heads.


*Typification: Ascopolyporus polychrous.*


*Habit and type host*: On dead culms of bamboo, stems or leaf midrib of monocotyledonous and dicotyledonous plants.

*Distribution*: Argentina, Bolivia, Brazil, Colombia, Costa Rica, Ecuador, Peru, Thailand [23].

***Ascopolyporus albus*** Mongkolsamrit, Thanakitpipattana and Luangsa-ard **sp. nov**. Figure 2.

*MycoBank*: MB 841855.

*Etymology*: From the Latin “albus”, referring to the white color of the fresh stromata.

*Typification*: Thailand, Chiang Mai Province, Chiang Dao Wildlife Sanctuary, Doi Chiang Dao Wildlife Research Station; 19°23′10.70″ N, 98°50′28.50″ E, on scale insects (*Coccidae; Hemiptera*), on living stem of bamboo (*Bambusae*), 17 August 2011, K. Tasanathai (K.T.), P. Srikitikulchai (P.S.), S. Mongkolsamrit (S.M.), A. Khonsanit (A.K.) (holotype BBH30734, ex-holotype culture BCC48975). GenBank: ITS = OL331502, LSU = OL322048, *TEF1* = OL322035, *RPB1* = OL322056, *RPB2* = OL322065.

*Description*: *Stromata* epibiotic, pulvinate, subglobose, globose, white (NN155C) to pinkish white (N155B), becoming dark brown when old, 3–6 mm wide, 2–3 mm thick. *Perithecia* semi-immersed, with slightly protruding orifices, obpyriform, 250–320 × 100–120 µm. *Asci* cylindrical up to 250 µm long and 3–4 µm wide, *Asci caps* 2–3 × 3–4 μm. *Ascospores* hyaline, whole, filiform, multiseptate, 95–135 × 1 µm.

*Culture characteristics*: Colonies on OA attaining a diameter of 3.5–4 cm in 14 days, slightly convex to the agar surface, white, reverse moderate orange yellow (164C). *Phialides* arising from aerial hyphae, solitary, cylindrical, slightly curved, up to 80 µm long, 1–2 µm wide. *Conidia* hyaline, enteroblastic, fusiform to acerose, early in development aseptate, becoming 1–4 septa, aggregated at the apex of the phialides, (8–)10–23(–28) × (2–)2.5–3 µm. Colonies on PDA attaining a diameter of 3.5–4 cm in 14 days, slightly convex to the agar surface, white, reverse light yellow (162C). *Phialides* arising from aerial hyphae, solitary, cylindrical, slightly curved, up to 120 µm long, 1–2 µm wide. *Conidia* hyaline, enteroblastic, fusiform to acerose, early in development aseptate, developing 1–4 septa, aggregated at the apex of the phialides, (8–)10.5–21(–30) × (2–)2.5–3(–3.5) µm.

*Habitat*: On scale insects (*Coccidae*; *Hemiptera*), found on living stems of bamboo (*Bambusae*).

*Additional specimen examined*: Thailand, Chiang Mai Province, Chiang Dao Wildlife Sanctuary, Doi Chiang Dao Wildlife Research Station; 19°23′10.70″ N, 98°50′28.50″ E, on scale insects (*Coccidae*; *Hemiptera*), on the living stems, 17 August 2011, K.T., P.S., S.M., A.K. (BBH30734, BCC48976). GenBank: ITS = OL331503, LSU = OL322049, *TEF1* = OL322036, *RPB1* = OL322057, *RPB2* = OL322066.

*Notes*: *Ascopolyporus albus* significantly differs from other species in *Ascopolyporus* herein. The difference is in the color of stromata. *Ascopolyporus albus* produces white to pinkish white stromata (Figure 2), whereas other species produce very pale violet (91D) to yellowish white (158) with strong orange (25A) stromata. Based on *Ascopolyporus* species from Thailand, the perithecia of *A. albus* are semi-immersed, similar to those in *A. galloides*, *A. griseoperitheciatus* and *A. purpuratus*. The perithecial shape of *A. albus* differs from *A. galloides*, *A. griseoperitheciatus* and *A. purpuratus* by having an obpyriform shape, whereas perithecia in *A. galloides*, *A. griseoperitheciatus* and *A. purpuratus* are obclavate, obovoid and ovoid, respectively.

***Ascopolyporus caulium*** (Berk. and M.A. Curtis) Thanakitp. and Luangsa-ard, **comb. nov**.

*MycoBank*: MB 842779.

 ≡ *Corticium caulium* Berk. and M.A. Curtis, J. Acad. nat. Sci. Philad. 2: 279. 1854.

 ≡ *Hypocrella caulium* (Berk. and M.A. Curtis) Pat., Bull. Soc. Mycol. France 30: 346. 1915.

 ≡ *Hyperdermium caulium* (Berk. and M.A. Curtis) P. Chaverri and K.T. Hodge, 2008.

= *Hypocrella camerunensis* Henn., Engler’s Bot. Jahrb. 23: 540. 1897.

= *Hypocrella brasiliana* (Henn.) Mains, Mycopath. Myc. Appl. 11: 311. 1959.

 ≡ *Stigmatea brasiliana* Henn., Hedwigia 36: 230. 1897.

 ≡ *Hypocrella camerunensis* var. *brasiliana* Henn., Hedwigia 43: 85. 1904.

= *Hyperdermium bertonii* J.F. White, R.F. Sullivan, Bills and Hywel-Jones, Mycologia 92: 910. 2000.

 ≡ *Epichloë bertonii* Speg., An. Mus. Nac. Hist. Nat. Buenos Aires 31: 416. 1922.

***Ascopolyporus galloides*** Khonsanit, Thanakitpipattana and Luangsa-ard **sp. nov.**
Figure 3.

*MycoBank*: MB 841853.

*Etymology*: Refers to the character of the stromata, which look similar to plant galls.

*Typification*: Thailand, Nakhon Ratchasima Province, Khao Yai National Park, Mo Singto Nature Trail; 14°26′21.46″ N, 101°22′20.20 ″E, on scale insects (*Coccidae; Hemiptera*), on the living stems of dicotyledonous plant, 5 July 2011, A.K., K.T., K. Sansatchanon (K.S.), P.S., S.M., W. Noisripoom (W.N.) (holotype BBH30629, ex-holotype culture BCC48704). GenBank: ITS = OL331509, LSU = OL322044, *TEF1* = OL322031, *RPB1* = OL322055, *RPB2* = OL322062.

*Description*: *Stromata* epibiotic, pulvinate, subglobose, hemispherical, upper surface white (NN155B) to yellowish white (158); lower surface strong orange (N25C), 1–7 mm wide. *Perithecia* semi-immersed, crowded, obclavate, 170–340 × 60–110 μm. *Asci* cylindrical, (129–)133–153.5(–175) × (3–)3.5–5(–6) μm. *Asci caps* 1–2 × 2.5–4 μm. *Ascospores* hyaline, whole, filiform, aseptate, (131–) 154.5–211.5(–216) × 0.5 μm.

*Culture characteristics*: Colony on OA attaining a diameter of 3.5–4 cm in 20 days, flat, slightly convex to the agar surface, white (158), reverse pale greenish yellow (2D). *Phialides* arising from aerial hyphae, solitary, cylindrical, slightly curved, 35–161 × 1–2 μm. *Conidia* hyaline, enteroblastic, fusiform to acerose, early in development aseptate, becoming 1–3 septa, aggregated at the apex of the phialides, (4–)6–22(–34) × (1.5–)2–2.5(–3) µm.

Colony on PDA attaining a diameter of 4 cm in 20 days, fluffy in the middle, flat to umbonate, white in the middle, pale orange yellow (23D), light yellow (11B), brilliant yellow (11A), reverse moderate orange (173C) in the middle, strong orange yellow (N163D), light yellow (14D). *Phialides* arising from aerial hyphae, solitary, cylindrical, slightly curved, 30–294 × 1–2 μm. *Conidia* hyaline, enteroblastic, fusiform to acerose, cylindrical, early in development aseptate, becoming 1–4 septa, aggregated at the apex of the phialides, (5–)8–16(–27) × (2–)2.5–3.5(–4) µm.

*Habitat*: On scale insects (*Coccidae*, *Hemiptera*), found on living stems of dicotyledonous plant.

*Additional specimens examined*: Thailand, Ranong Province, Khlong Nakha Wildlife Sanctuary, Khlong Nakha Nature Trail; 9°27′33″ N, 98°30′16″ E, on scale insects (*Coccidae; Hemiptera*), on living stems of dicotyledonous plant, 5 October 2004, B. Thongnuch (B.T.), D. Johnson (D.J.), K.T., S.M., W. Chaygate (W.C.) (BBH10163, BCC16408; BBH10176, BCC16419; BBH10177, BCC16420; BBH10178, BCC16421; BBH10179, BCC16422; BBH10180, BCC16423); Nakhon Nayok Province, Khao Yai National Park, Tat Ta Phu Waterfall Nature Trail; 14°26′21.46” N, 101°22′20.20” E, on scale insects (*Coccidae*; *Hemiptera*), on living stems of dicotyledonous plant, 24 August 2005, K.T. (BBH14835, BCC18980); Phetchaburi Province, Kaeng Krachan National Park, Ban Krang Camp Nature Trail; 12°54′05″ N, 99°37′48″ E, on scale insects (*Coccidae*; *Hemiptera*), on living stems of dicotyledonous plant, 14 November 2005, B.T., K.T., R. Ridkaew (R.R.), W.C. (BBH15034, BCC19720); Ranong Province, Khlong Nakha Wildlife Sanctuary, Khlong Nakha Nature Trail; 9°27′33″ N, 98°30′16″ E, on scale insects (*Coccidae*; *Hemiptera*), on living stems of dicotyledonous plant, 10 January 2006, B.T., K.T., L.N. Yen (L.N.Y.), L.T. Huyen (L.T.H.), PS, SM, WC (BBH16500, BCC20115); 12 January 2006, B.T., K.T., L.N.Y., L.T.H., P.S., S.M., W.C. (BBH16554, BCC20123); Nakhon Ratchasima Province, Khao Yai National Park, Bueng Phai Nature Trail; 14°26′21.46″ N, 101°22′20.20″ E, on scale insects (*Coccidae*; *Hemiptera*), on living stems of dicotyledonous plant, 5 July 2006, B.T., J. Luangsa-ard (J.J.L.), K.T., P.S., S.M., W.C. (BBH18631, BCC22237; BBH18632, BCC22238); Chanthaburi Province, Khao Soi Dao Wildlife Sanctuary, Withiphrai Nature Trail; 13°06′13″ N, 102°11′39″ E, on scale insects (*Coccidae*; *Hemiptera*), on living stems of dicotyledonous plant, 1 May 2007, B.T., K.T., R.R., S.M., W.C. (BBH19873, BCC25446; BCC25447, BCC25448); Nakhon Ratchasima Province, Khao Yai National Park, km. 33 Nature Trail; 14°26′21.46″ N, 101°22′20.20″ E, on scale insects (*Coccidae*; *Hemiptera*), on living stems of dicotyledonous plant, 8 August 2007, B.T., P. Puyngain (P.P.), W.C. (BBH22627, BCC26680), Trang Province, Khao Chong Wildlife Development and Conservation Promotion Station, 1.8 km. Nature Trail; 7°32′57″ N, 99°47′11″ E, on scale insects (*Coccidae*; *Hemiptera*), on living stems of dicotyledonous plant, 18 September 2007, B.T., K.T. (BBH23089, BCC27812); Nakhon Ratchasima Province, Khao Yai National Park, Mo Singto Nature Trail; 14°26′21.46″ N, 101°22′20.20″ E, on scale insects (*Coccidae*; *Hemiptera*), on living stems of dicotyledonous plant, 18 June 2009, K.T., P.S., R.R., S.M., T. Chohmee (T.C.) (BBH30139, BCC36656); 20 July 2009, K.T., P.S., R.R., S.M., T.C. (BBH27634, BCC37668); 23 July 2009, K.T., P.S., R.R., S.M. (BCC37879); Khao Yai National Park, km. 29 Nature Trail; 14°26′21.46″ N, 101°22′20.20″ E, on scale insects (*Coccidae*; *Hemiptera*), on living stems of dicotyledonous plant, 2 June 2011, A.K., K.T., K.S., P.S., S.M., W.N. (BBH30577, BCC47981); GenBank: ITS = OL331511, LSU = OL322043, *TEF1* = OL322030, *RPB1* = OL322054, *RPB2* = OL322061; Khao Yai National Park, Mo Singto Nature Trail; 14°26′21.46″ N, 101°22′20.20″ E, on scale insects (*Coccidae; Hemiptera*), on living stems of dicotyledonous plant, 3 August 2011, A.K., K.T., K.S., P.S., S.M., W.N. (BBH30683, BCC48951).

*Notes*: Based on the macromorphologies of the natural samples, the lower surface of stromata of *A. galloides* and *A. griseoperitheciatus* are orange and their perithecial layers are white and pale violet to light purplish gray, respectively. The perithecia in these two species are semi-immersed, but perithecia in *A. galloides* are obclavate, whereas those of *A. griseoperitheciatus* are obovoid. The colony color of *A. galloides* on PDA is pale orange yellow, light yellow and brilliant yellow, whereas *A. griseoperitheciatus* is white and produces a pale purplish pink pigment diffusing in the medium.

***Ascopolyporus griseoperitheciatus*** Khonsanit, Thanakitpipattana and Luangsa-ard **sp. nov**. Figure 4.

*MycoBank*: MB 841854.

*Etymology*: From the Latin “griseo”, referring to the gray color of the fresh stromata.

*Typification*: Thailand, Kamphaeng Phet Province, Khlong Lan National Park, Khlong Lan Waterfall; 16°07′50.20″ N, 99°16′36.30″ E, on scale insects (*Coccidae*; *Hemiptera*), on the living stems of dicotyledonous plant, 19 June 2006, B.T., J.J.L., K.T., P.S., R.R., S.M., W.C. (holotype BBH18679, ex-holotype culture BCC22358). GenBank: ITS = OL331507, LSU = OL322050, *TEF1* = OL322037, *RPB1* = *RPB2* = OL322067. 

*Description*: *Stromata* epibiotic, irregularly pulvinate or subglobose, upper surface very pale violet (91D) to light purplish gray (N187D); lower surface vivid yellow (14C) to strong orange (25A), 3–7 mm wide. *Perithecia* semi-immersed, crowded, obovoid, 150–320 × 80–140 μm. *Asci* cylindrical, (150–)154–179(–193) × 4–5 μm. *Asci caps* 1.5–2 × 3–3.5 μm. *Ascospores* hyaline, whole, filiform, aseptate, extending the length of ascus.

*Culture characteristics*: Colony on OA attaining a diameter of 4 cm in 20 days, flat, slightly convex to the agar surface, white with light yellow green (150D), reverse pale orange yellow (23D). *Phialides* arising from aerial hyphae, solitary, cylindrical or acremonium-like, slightly curved, 50–250 × 1–2 μm. *Conidia* hyaline, enteroblastic, fusiform to acerose, early in development aseptate, becoming 1–2 septa, aggregated at the apex of the phialides, (6–)7–14(–19) × (1.5–)2–3 µm.

Colony on PDA attaining a diameter of 3.5–4 cm in 20 days, compact mycelium, slightly convex to the agar surface, white (158), pale purplish pink (56A) pigment diffusing in medium, reverse strong yellowish pink (31C). *Phialides* arising from aerial hyphae, solitary, cylindrical or acremonium-like, slightly curved, 43–265 × 1–2 μm. *Conidia* hyaline, enteroblastic, fusiform to acerose, early in development aseptate, becoming 1–2 septa, aggregated at the apex of the phialides, (4–)6–11(–17) × 2–3.5(–4) µm. 

*Habitat*: On scale insects (*Coccidae*; *Hemiptera*), found on living stems of dicotyledonous plants.

*Additional specimens examined*: Thailand, Chanthaburi Province, Khao Soi Dao Wildlife Sanctuary, Withiphrai Nature Trail; 13°06′13″ N, 102°11′39″ E, on scale insects (*Coccidae*; *Hemiptera*), on living stems of dicotyledonous plant, 1 May 2007, B.T., K.T., R.R., S.M., W.C. (BBH19872, BCC25788); Nakhon Ratchasima Province, Khao Yai National Park, Mo Singto Nature Trail; 14°26′21.46″ N, 101°22′20.20″ E, on scale insects (*Coccidae*; *Hemiptera*), on living stems of dicotyledonous plant, 30 June 2010, A.K., K.T., K.S., P.S., R. Somnuk (R.S.), S.M. (BBH30155, BCC43315).

*Notes*: Our molecular phylogenetic study has shown that *A. griseoperitheciatus* is closely related to *A. albus*. However, *A. griseoperitheciatus* significantly differs from *A. albus* in having a perithecial layer on the upper surface of stromata that is very pale violet to light purplish gray, the lower surface of stromata is vivid yellow to strong orange, while in *A. albus*, the stromata are only white. Additionally, *A. griseoperitheciatus* produces a pale purplish pink pigment diffusing in PDA plates, whereas *A. albus* does not produce any pigment.

***Ascopolyporus gollmerianus*** Henn., Hedwigia 41: 8. 1902.

***Ascopolyporus khaoyaiensis*** Mongkolsamrit, Thanakitpipattana and Luangsa-ard sp. nov. Figure 5.

*MycoBank*: MB 841856.

*Etymology*: Named after Khao Yai National Park, where the type specimen was found.

*Typification*: Thailand, Nakhon Ratchasima Province, Khao Yai National Park, Mo Singto Nature Trail; 14°26′21.46″ N, 101°22′20.20″ E, on scale insects (*Coccidae*; *Hemiptera*), on the living stems of dicotyledonous plant, 5 A0ugust 2010, K.T., P.S., S.M., A.K., R.S., K.S. (holotype BBH30157, ex-holotype culture BCC43741). GenBank: ITS = OL331513, LSU = OL322041, *TEF1* = OL322040, *RPB2* = OL322070. 

*Description*: *Stromata* epibiotic, flattened to convex, cylindrical to irregularly shaped, upper surface very pale violet (91D) to dark purple (59A); lower surface white to pale orange (20A), 3–25 mm wide, 1–3 mm thick. *Perithecia* semi-immersed, slightly protruding apices, narrow flask shaped, slightly protruding, obclavate, 300–360 × 100–120 µm. *Asci* cylindrical, up to 215 µm long, 3–4 µm wide, *Asci caps* 2–4 × 3–4 μm. *Ascospores* hyaline, whole, filiform, aseptate, 175–200 × 1 µm.

*Culture characteristics*: Colonies on OA attaining a diameter of 3.5 cm in 14 days, cottony, white, reverse moderate brown (165A). *Phialides* arising from aerial hyphae, solitary, cylindrical or acremonium-like, slightly curved, up to 60 µm long, 1–2 µm wide. *Conidia* hyaline, enteroblastic, fusiform to acerose, early in development aseptate, mostly becoming 1 septum, occasionally 2–3 septa, aggregated at the apex of the phialides, (5–)8–16(–20) × (1.5–)2–3 µm.

Colonies on PDA attaining a diameter of 3–4 cm in 14 days, cottony, white, pale orang in the middle of colony, reverse dark red (59A) in the middle, bright strong purplish red (60D) pigment diffusing in medium. *Phialides* arising from aerial hyphae, solitary, cylindrical or acremonium-like, up to 50 µm long, 1–2 µm wide. *Conidia* hyaline, enteroblastic, fusiform to acerose, early in development aseptate, mostly becoming 1 septum, occasionally 2–3 septa, aggregated at the apex of the phialides, (7–)9–16.5(–22) × (1.5–)2–3 µm.

*Habitat*: On scale insects (*Coccidae*; *Hemiptera*), found on living stems of dicotyledonous plant.

*Additional specimen examined*: Thailand, Nakhon Ratchasima Province, Khao Yai National Park, Mo Singto Nature Trail; 14°26′21.46″ N, 101°22′20.20″ E, on scale insects (*Coccidae*; *Hemiptera*), on living stems of dicotyledonous plant, 30 June 2010, K.T., P.S., S.M., A.K., R.S., K.S. (BBH30154, BCC43314). GenBank: ITS = OL331512, LSU = OL322052, *TEF1* = OL322039, *RPB2* = OL322069.

*Notes*: The stromatal color of the natural samples of *A. khaoyaiensis* is similar to the purple stromata of *A. purpuratus.* However, perithecia in *A. khaoyaiensis* are immersed and obclavate, whereas perithecia in *A. purpuratus* are semi-immersed and ovoid. Asci of *A. khaoyaiensis* are shorter than those of *A. purpuratus* (up to 215 × 3–4 vs. 200–240 × 4–5 µm). Additionally, *A. khaoyaiensis* and *A. purpuratus* produce bright strong purplish red pigment diffusing in PDA plates.

***Ascopolyporus möellerianus*** (Henn.) Möller, Phycomyc. Ascomyc. Bras.: 301. 1901.

***Ascopolyporus philodendri*** J.F. Bisch. (as “philodendrus”), Mycologia 97(3): 711. 2005.

***Ascopolyporus polychrous*** Möller, Bot. Mitt. Trop. 9: 300. 1901.

***Ascopolyporus polyporoïdes*** Möller, Bot. Mitt. Trop. 9: 301. 1901.

***Ascopolyporus purpuratus*** Mongkolsamrit, Thanakitpipattana, Himaman and Luangsa-ard **sp.**
**nov**. Figure 6.

*MycoBank*: MB 841857.

*Etymology*: Referring to the purple color of the fresh stroma.

*Typification*: Thailand, Nakhon Ratchasima Province, Khao Yai National Park, Pong Chang Chomrom Phoen (Nong Phakchi); 14°27’04.0″ N, 101°22’03.60″ E, on scale insects (*Coccidae*; *Hemiptera*), on the living stems and midrib of dicotyledonous leaves, 19 September 2018, J.J.L., K.T., D. Thanakitpipattana (D.T.), B. Sakolrak (B.S.), R.S., S.M., W.N., W. Himaman (W.H.), P.S. (holotype BBH44511, ex-holotype culture BCC88430). GenBank: ITS = OL331506, LSU = OL322045, *TEF1* = OL322032, *RPB1* = OL322059.

*Description*: *Stromata* epibiotic, flattened to convex, consisting of dense white mycelial mat, upper surface yellow (18D) to very pale purple-violet (75D), 5–12 mm long, 3–8 mm wide. *Perithecia* semi-immersed, crowded, ovoid, (300–)335–414(–420) × (100–)110–142(–150) µm. *Asci* cylindrical, up to 240 × 2–4 µm. *Asci caps* 2–4 × 3–4 μm. *Ascospores* hyaline, whole, filiform, aseptate, (100–)131–190(–220) × 1–1.5 µm.

*Culture characteristics*: Colonies on OA attaining a diameter of 3–4 cm in 14 days, flat, white, reverse white. *Phialides* arising from aerial hyphae, solitary, cylindrical or acremonium-like, slightly curved, up to 40 µm long, 1–2 µm wide. *Conidia* hyaline, enteroblastic, fusiform to acerose, aseptate, aggregated at the apex of the phialides, (5–)6.5–18(–25) × (1.5–)2–2.5(–3) µm.

Colonies on PDA attaining a diameter of 2.5–3.5 cm in 14 days, slightly convex to the agar surface, vivid orange (28B) with white in the middle of colony, reverse strong reddish orange (34C), bright moderate reddish orange (N34D) pigment diffusing in medium. *Phialides* arising from aerial hyphae, solitary, cylindrical or acremonium-like, slightly curved, up to 55 µm long, 1–2 µm wide. *Conidia* hyaline, enteroblastic, fusiform to acerose, aseptate, aggregated at the apex of the phialides, (5–)7–13.5(–18) × 1.5–2.5(–3) µm.

*Habitat*: On scale insects (*Coccidae; Hemiptera*), found on living stem and midrib of leaves of dicotyledonous plants.

*Additional specimen examined*: Thailand, Phetchaburi Province, Kaeng Krachan National Park, Ban Krang Camp Nature Trail; 12°54′05″ N, 99°37′48″ E, on scale insects (*Coccidae; Hemiptera*), on the midrib of leaves, 14 November 2005, K.T., W.C., R.R., B.T. (BBH 15035, BCC 19721); Nakhon Ratchasima Province, Khao Yai National Park, Mo Singto Nature Trail; 14°26′21.46″ N, 101°22′20.20″ E, on scale insects (*Coccidae; Hemiptera*), on the twigs of tree, 23 July 2009, K.T., P.S., R.R., S.M. (BBH26373, BCC37880); Nakhon Ratchasima Province, Khao Yai National Park, Bueng Phai Nature Trail; 14°26′21.46″ N, 101°22′20.20″ E, on scale insects (*Coccidae*; *Hemiptera*), on the midrib of leaves and twigs of tree, 18 September 2018, J.J.L., K.T., D.T., B.S., R.S., S.M., W.N., W.H., P.S. (BBH44547, BCC88388); GenBank: ITS = OL331505, LSU = OL322046, *TEF1* = OL322033, RPB2 = OL322064, (BBH44551, BCC88389); GenBank: ITS = OL331504, LSU = OL322047, *TEF1* = OL322034.

*Notes*: *Ascopolyporus purpuratus* can be found on the midrib of leaves and living stems of dicotyledonous plants. In natural samples, the perithecia are pale yellow and/or very pale purple-violet. Additionally, *A. purpuratus* produces red pigment diffusing in PDA plates the same as *A. griseoperitheciatus* and *A. khaoyaiensis*. However, the mycelia of *A. purpuratus* are white in the middle of the colony with orange edges, whereas *A. griseoperitheciatus* produces only white mycelia and *A. khaoyaiensis* produces white mycelia that turn pale orange in the center of colony.

***Ascopolyporus villosus*** Möller, Bot. Mitt. Trop. 9: 301. 1901.

***Neohyperdermium*** Thanakitpipattana and Luangsa-ard, gen. nov.

*MycoBank*: MB 842780.

*Etymology*: Referring to the phenotypic similarity of the stromatal formation to *Hyperdermium*.

*Typification*: *Neohyperdermium piperis* (J.F. Bisch. and J.F. White) Thanakitpipattana and Luangsa-ard.

*Description*: *Stroma* epibiotic, flattened to pulvinate, white to yellow. Hosts are scale insects (*Coccoidea*, *Hemiptera*). *Perithecia* immersed, obpyriform, cymbiform to cone-shaped. *Asci* cylindrical, linear with enlarged refractive tip. Asexual morph verticillium-like.

*Notes*: This genus is a phylogenetically separate lineage from other scale insect pathogens in *Cordycipitaceae*, as shown in Figure 1. Two species are recognized in this genus that produce white to yellow stromata, immersed perithecia and a verticillium-like anamorph.

***Neohyperdermium piperis*** (J.F. Bisch. and J.F. White) Thanakitpipattana and Luangsa-ard, **comb. nov**.

MycoBank MB 842782.

*≡ Torrubiella piperis* J.F. Bisch. and J.F. White, Studies in Mycology 50: 89–94. 2004.

*≡ Cordyceps piperis* (J.F. Bisch. and J.F. White) D. Johnson, G.H. Sung, J.F. Bisch. and Spatafora, Mycol. Res. 113(3): 284. 2009.

*Description and illustration*: See J.F. Bisch. and J.F. White (2004).

*Typification*: Panama, Barro Colorado Island, Lutz Creek, scale insect (*Coccoidea*, *Hemiptera*) on *Piper carrilloanum* (*Piperaceae*) August 2003, J.F. Bischoff and J.F. White, Jr., New York Botanical Garden (NY), culture ex-type CBS 116719.

*Habitat*: Scale insects.

Known distribution: Panama.

*Note*: *Neohyperdermium piperis* is closely related to *N. pulvinatum* and can be distinguished from *N. pulvinatum* in producing part-ascospores, whereas in *N. piperis*, the ascospores are whole with multiple septations and the conidia are aseptate.

***Neohyperdermium pulvinatum*** (J.F. White et al.) Thanakitpipattana and Luangsa-ard, comb. nov.

*MycoBank*: MB 842783.

*≡ Hyperdermium pulvinatum* J.F. White et al., Mycologia 92(5): 908–918. 2000.

*Description and illustration*: See J.F. White et al. (2000).

*Typification*: Costa Rica, Guanacaste, Parque Nacional Guanacaste, Sector El Hacha, Puesto Los Almendros, on *Asteraceae*, 6 October 1998, J.F. White, G. Bills and S. Salas, RUTPP, culture ex-type ATCC MYA-69.

*Habitat*: Scale insects.

Known distribution: Costa Rica.

*Note*: *Neohyperdermium pulvinatum* is closely related to *N. piperis*, which can be distinguished by the type of ascospores and the presence of multiseptate conidia.

## 4. Discussion

The results of our multigene phylogenetic analyses show that our specimens were closely related to *Ascopolyporus polychorus*, *A. villosus* and *Hyperdermium caulium* (Figure 1). Importantly, the specimens in this study are clearly distinct species in *Ascopolyporus* because of the differences in the sizes, color, perithecial position and features of the stromata, which also overlap with morphological characters of some species previously treated as belonging to the genera *Hyperdermium* (*H. caulium*, *H. bertonii*, *H. pulvinatum*) and *Cordyceps* (*C. piperis*) in *Cordycipitaceae*, by producing flattened to pulvinate stromata and producing unique cylindrocarpon-like anamorph with multiseptate conidia [11,12,14,16]. The two genera, *Ascopolyporus* and *Hyperdermium*, differ only in the sizes and characters of ascomata [11,13,14]; *Hyperdermium* stromata are either flattened or pulvinate, whereas in *Ascopolyporus* sensu Möller, the stromata are subglobose to polypore-like. Based on these results, since the type species of *Hyperdermium*, *H. caulium*, is nested within *Ascopolyporus*, *Hyperdermium* is synonymized with *Ascopolyporus* and a new combination is proposed for *H. caulium*. The generic description of *Ascopolyporus* is therefore emended to include flattened to pulvinate stromata.

Our new species in *Ascopolyporus* are characterized by possessing flattened and pulvinate stromata, two groups that are supported as separate clades in phylogenetic analyses (Figure 1). The three pulvinate species (*Ascopolyporus albus*, *A. galloides* and *A. griseoperitheciatus*) have smaller stromata than previously described by Möller [24] and Bischoff et al. [14]. The two new species of *Ascopolyporus khaoyaiensis* and *A. purpuratus* have flattened stromata, and their sizes are in the same range as *A. caulium* (Table 2). All new *Ascopolyporus* species in this study possess semi-immersed perithecia with ostioles slightly protruding on the surface of the fertile cushion, whereas *A. polychrous* and *A. philodendrus* have completely immersed perithecia; *A. vilosus* does not produce perithecia on stromata [14].

Another species in *Hyperdermium* found in *Cordycipitaceae*, *H. pulvinatum*, did not cluster with type species *H. caulium*, which was congruent with Sung et al. [45], Kepler et al. [44] and Wang et al. [54], and is grouped together with *Cordyceps piperis* possessing pulvinate stromata that are white to yellow, producing aseptate, subcylindrical conidia on cultures (Table 2). These two species are proposed as new combinations in a new genus *Neohyperdermium*, as *Neohyperdermium pulvinatum* and *N. piperis*, which were described as epiphytes on scale insect pathogens in *Cordycipitaceae*.

The evolution and ecology of insect pathogenic fungi using insects and plants as the main source of nutrients remain not fully understood. Humber [55] suggested that the interaction between higher fungi and plants range from virulent pathogens to decomposer to mutualistic symbiosis. In *Hypocreales*, the genera *Aschersonia*, *Ascopolyporus*, *Conoideocrella*, *Dussiella*, *Hyperdermium*, *Hypocrella*, *Moelleriella, Regiocrella* and *Samuelsia* also utilize nutrients from the phloem of host plants through scale insects and white flies (*Coccidae* and *Aleyrodidae*) to continue their growth on plants [11,13,14,15]. Our new *Ascopolyporus* species are also found in this position, in which the scale insect attached to host plants was parasitized until it was consumed, but the fungus continues to utilize the nutrients that are being released through the stylet apparatus. The interactions occurred on the underside of fungal stroma, which is where the bridge for the exchange of nutrients between the fungus and plant exists (Figure 7).

Hypocrealean fungi are excellent producers of secondary metabolites which can be used to reduce the damage from insect fungi herbivores and phytopathogenic fungi [13,56]. However, no report has been made on the secondary metabolites produced from any of the reported species in *Ascopolyporus*, which should be a focus of future studies.

**Residual Species of *Ascopolyporus***.

The remaining taxon could not be accommodated in the genus *Ascopolyporus* because its morphological description resembles other genera in *Clavicipitaceae* by producing paraphyses, which are not found in *Ascopolyporus* (*Cordycipitaceae*; *Hypocreales*), and molecular phylogenetic data are not available.

***Ascopolyporus puttemansii*** Henn., Hedwigia 48: 6. 1908.


**Key to *Ascopolyporus* species:**


1a. Conidia aseptate……………………………………………………………………………………………2

1b. Conidia aseptate to multiseptate…………………………………………………………………………3

2a. Conidia oval………………………………………………………………………………………………..*A. möllerianus*

2b. Conidia fusiform to acerose………………………………………………………………………………*A. purpuratus*

3a. Conidia 1–5 septate, cylindrical to fusiform, 5–30 × 1–3 µm………………………………………….*A. caulium*

3b. Conidia 1–4 septate………………………………………………………………………………………..4

4a. Conidia subcylindrical…………………………………………………………………………………….5

4b. Conidia fusiform to acerose………………………………………………………………………………6

5a. Conidia subcylindrical, 7–25 × 3–4 µm………………………………………………………………….*A. philodendrous*

5b. Conidia subcylindrical, guttulate, 10–22 × 2–5 µm……………………………………………………*A. villosus*

6a. Ascospores hyaline, disarticulate into part-spores…………………………………………………….7

6b. Ascospores hyaline, whole……………………………………………………………………………….8

7a. Ascospores filiform to spiroid, 6 × 1 µm……………………………………………………………….*A. polychrous*

7b. Ascospores filiform, 8–15 µm……………………………………………………………………………*A. polyporoïdes*

8a. Ascospores multiseptate, 95–135 × 1 µm……………………………………………………………….*A. albus*

8b. Ascospores aseptate……………………………………………………………………………………….9

9a. Perithecia semi-immersed, obovoid…………………………………………………………………….*A. griseoperithciatus*

9b. Perithecia semi-immersed, obclavate…………………………………………………………………..10

10a. Stromata pulvinate, hemispherical, 1–7 mm…………………………………………………………*A. galloides*

10b. Stromata flattened to convex, cylindrical to irregular shaped, 3–25 mm…………………………*A. khaoyaiensis*


**Key to *Neohyperdermium* species**


1a. Ascospores filiform, multiseptate, whole………………………………………………………………*N. pulvinatum*

1b. Ascospores filiform, disarticulating into part-spores…………………………………………………*N. piperis*

## Figures and Tables

**Figure 1 jof-08-00516-f001:**
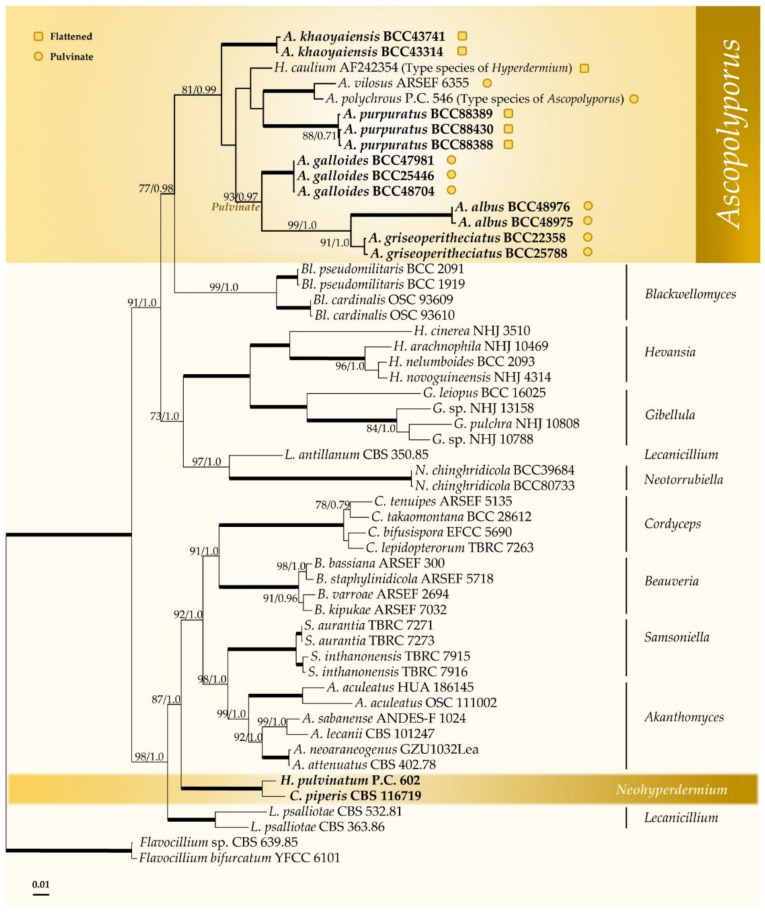
Phylogenetic reconstruction of *Ascopolyporus* and related genera in the *Cordycipitaceae* obtained from the combined LSU, *TEF1*, *RPB1* and *RPB2* sequence dataset based on maximum likelihood (RAxML) and Bayesian inference. Numbers on the nodes are ML bootstrap and Bayesian posterior probability values above 70% (MLBS) or 0.7 (BIPP). Thickened lines mean support for the two analyses was 100% (MLBS) or 1.0 (BIPP). 
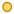
 represents species with pulvinate stromata while 
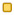
 represents species with flattened stromata.

**Figure 2 jof-08-00516-f002:**
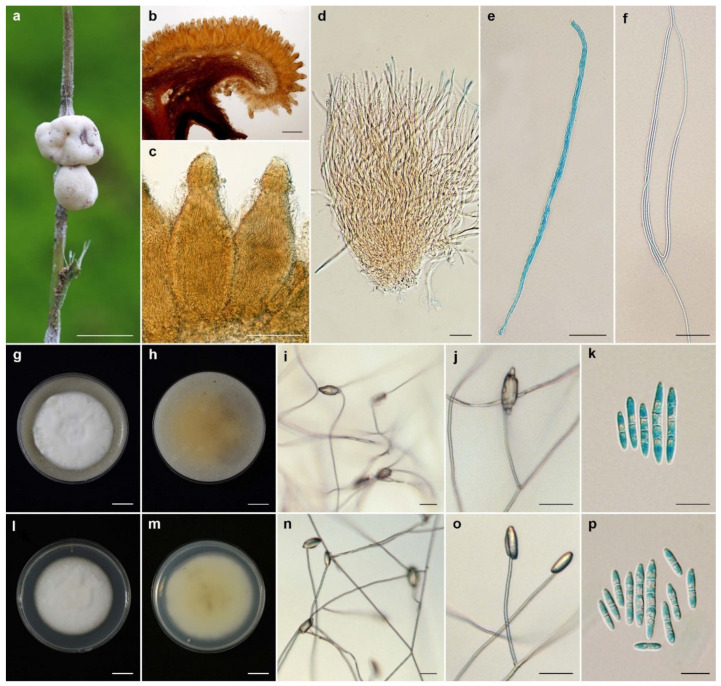
*Ascopolyporus albus*. (**a**) Stromata on living stem of bamboo (*Bambusae*); (**b**) cross-section through stroma showing perithecia (BBH30734); (**c**) perithecia; (**d**,**e**) asci. (**f**) ascospores; (**g**) colony obverse on OA; (**h**) colony reverse on OA; (**i**,**j**) phialide apex with conidial head on OA; (**k**) conidia on OA; (**l**) colony obverse on PDA; (**m**) colony reverse on PDA; (**n**,**o**) phialide and conidia on PDA; (**p**) conidia on PDA. Scale bars: (**g**,**h**,**l**,**m**) = 10 mm; (**a**) = 5 mm; (**b**) = 200 μm; (**c**) = 100 μm; (**d**,**f**,**j**,**n**,**o**) = 20 μm; (**e**,**k**,**p**) = 10 μm.

**Figure 3 jof-08-00516-f003:**
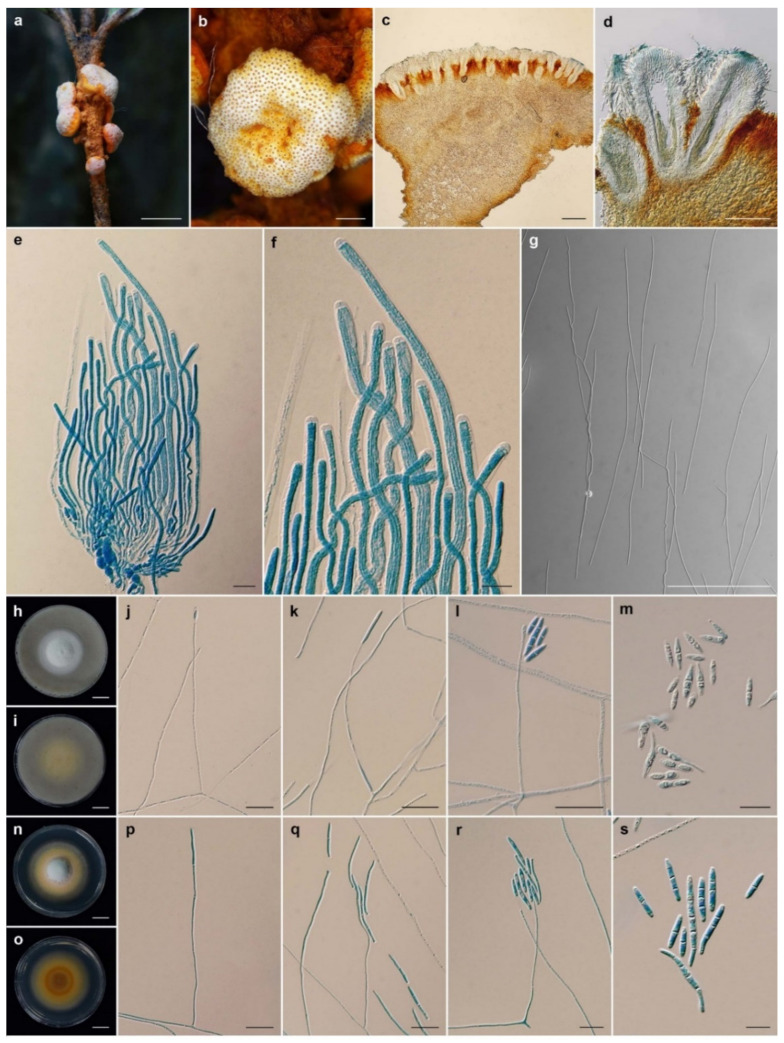
*Ascopolyporus galloides*. (**a**,**b**) Stromata on living stem of dicotyledonous plant (BBH48704); (**c**) cross-section through stroma showing perithecia; (**d**) perithecia; (**e**) asci; (**f**) asci-caps; (**g**) ascospores; (**h**) colony obverse on OA; (**i**) colony reverse on OA; (**j**–**l**) phialide apex with conidial head on OA; (**m**) conidia on OA; (**n**) colony obverse on PDA; (**o**) colony reverse on PDA; (**p**–**r**) phialide apex with conidial head on PDA; (**s**) conidia on PDA. Scale bars: (**h**,**i**,**n**,**o**) = 10 mm; (**a**) = 5 mm; (**b**) = 1 mm; (**c**) = 200 μm; (**d**,**g**) = 100 μm; (**j**,**k**,**l**,**q**,**r**) = 20 μm; (**e**,**f**,**m**,**p**,**s**) = 10 μm.

**Figure 4 jof-08-00516-f004:**
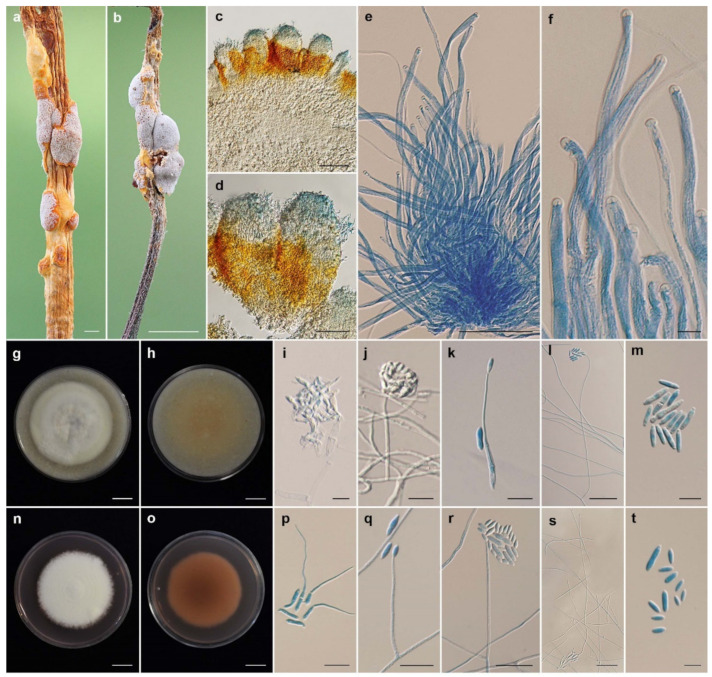
*Ascopolyporus griseoperitheciatus*. (**a**,**b**) Stromata on living stem of dicotyledonous plants (**a** BBH18679, **b** BBH30155); (**c**,**d**) perithecia; (**e**) asci; (**f**) asci-caps; (**g**) colony obverse on OA; (**h**) colony reverse on OA; (**i**) microsclerotium-like structure on OA; (**j**) phialide apex with conidial head; (**k**) conidium germination; (**l**) phialide and conidia on OA; (**m**) conidia on OA; (**n**) colony obverse on PDA; (**o**) colony reverse on PDA; (**p**) conidia germination; (**q**–**s**) phialide and conidia on PDA; (**t**) conidia on PDA. Scale bars: (**a**,**g**,**h**,**n**,**o**) = 10 mm; (**b**) = 5 mm; (**c**) = 100 μm; (**d**,**e**,**l**) = 50 μm; (**p**,**q**,**r**,**s**) = 20 μm; (**i**,**j**,**k**,**m**,**t**) = 10 μm; (**f**) = 5 μm.

**Figure 5 jof-08-00516-f005:**
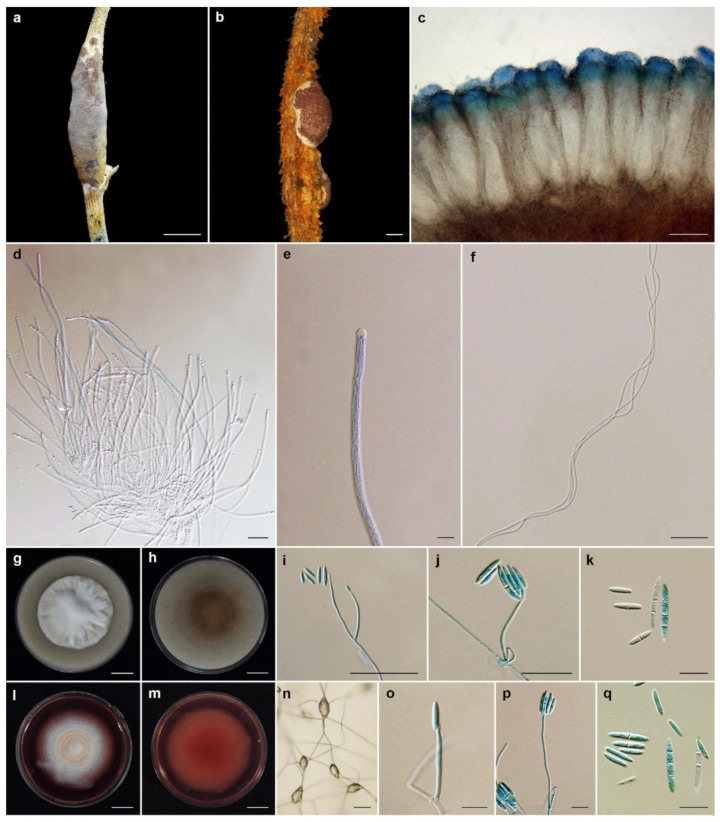
*Ascopolyporus khaoyaiensis*. (**a**,**b**) Stromata on living stem of dicotyledonous plants (**a** BBH30157, **b** BBH30154); (**c**) perithecia; (**d**) asci; (**e**) asci-caps; (**f**) ascospores; (**g**) colony obverse on OA; (**h**) colony reverse on OA; (**i**,**j**) phialide apex with conidial head on OA; (**k**) conidia on OA; (**l**) colony obverse on PDA; (**m**) colony reverse on PDA; (**n**–**p**) phialide apex with conidial head on PDA; (**q**) conidia on PDA. Scale bars: (**b**,**g**,**h**,**l**,**m**) = 10 mm; (**a**) = 5 mm; (**c**) = 100 μm; (**i**) = 50 μm; (**d**,**f**) = 20 μm; (**g**,**j**,**k**,**n**,**o**,**p**,**q**) = 10 μm; (**e**) = 5 μm.

**Figure 6 jof-08-00516-f006:**
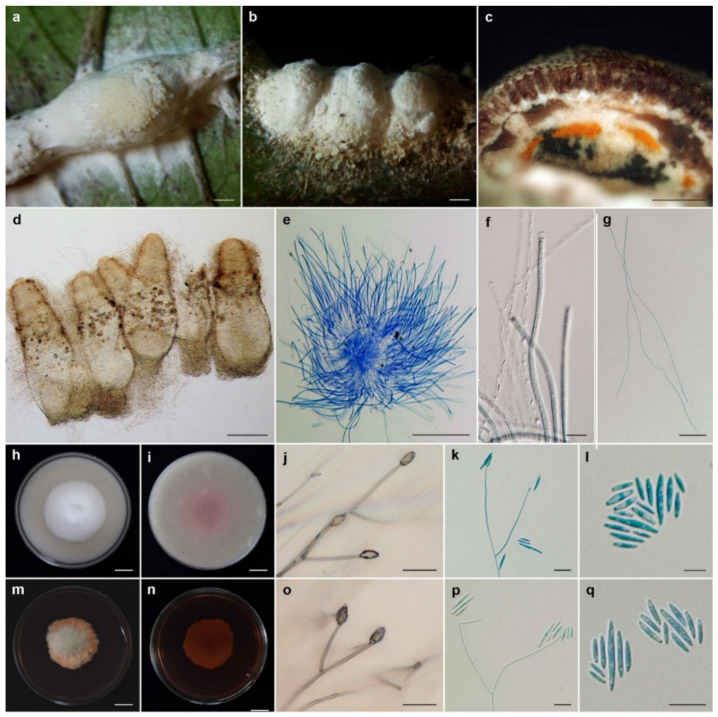
*Ascopolyporus purpuratus*. (**a**,**b**) Stromata on living midrib of leaves of dicotyledonous plant (BBH44511); (**c**) cross-section through stroma showing perithecia; (**d**) ovoid perithecia; (**e**) asci; (**f**) asci-caps; (**g**) ascospores; (**h**) colony obverse on OA; (**i**) colony reverse on OA; (**j**,**k**) phialide apex with conidial head on OA; (**l**) conidia on OA; (**m**) colony obverse on PDA; (**n**) colony reverse on PDA; (**o**,**p**) phialide apex with conidial head on PDA; (**q**) conidia on PDA. Scale bars: (**h**,**i**,**m**,**n**) = 10 mm; (**a**,**b**,**c**) = 1 mm; (**d**,**e**) = 100 μm; (**g**,**j**,**o**) = 20 μm; (**f**,**k**,**l**,**p**,**q**) = 10 μm.

**Figure 7 jof-08-00516-f007:**
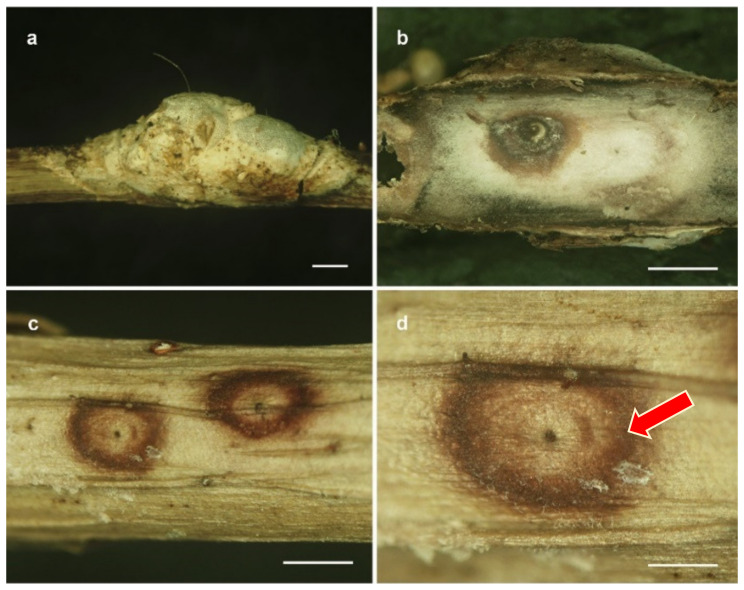
Photographs showing the interaction between fungus, plant and scale insect. (**a**) Fungal stroma on the stem of a dicotyledonous plant. (**b**) Underside of fungal stroma removed from the stem. (**c**,**d**) Underside of fungal stromata with ventage (arrow) from which the scale insect stylet entered the plant host. Scale bars = (**a**,**b**,**c**) = 1 mm; (**d**) = 0.05 mm.

**Table 1 jof-08-00516-t001:** List of species and GenBank accession numbers of sequences used in this study. Bold accession numbers were generated for this study. The symbol “–” denotes no available data.

Species	Strain	GenBank Accession No.				References
		LSU	*TEF1*	*RPB1*	*RPB2*	
*Akanthomyces aculeatus*	HUA 186145	MF416520	MF416465	–	–	[44]
*Akanthomyces attenuatus*	CBS 402.78	AF339565	EF468782	EF468888	EF468935	[45]
*Akanthomyces lecanii*	CBS 101247	AF339555	DQ522359	DQ522407	DQ522466	[45]
*Akanthomyces neoaraneogenus*	GZU1032Lea	KX845704	KX845698	KX845700	KX845702	[46]
*Akanthomyces sabanense*	ANDES-F 1024	KC875225	KC633266	–	KC633249	[47]
*Akanthomyces tuberculatus*	OSC 111002	DQ518767	DQ522338	DQ522384	DQ522435	[45]
** *Ascopolyporus albus* **	**BCC48975**	**OL322048**	**OL322035**	**OL322056**	**OL322065**	**This study**
** *Ascopolyporus albus* **	**BCC48976**	**OL322049**	**OL322036**	**OL322057**	**OL322066**	**This study**
** *Ascopolyporus galloides* **	**BCC25446**	**OL322042**	**OL322029**	**OL322053**	**OL322060**	**This study**
** *Ascopolyporus galloides* **	**BCC47981**	**OL322043**	**OL322030**	**OL322054**	**OL322061**	**This study**
** *Ascopolyporus galloides* **	**BCC48704**	**OL322044**	**OL322031**	**OL322055**	**OL322062**	**This study**
** *Ascopolyporus griseoperitheciatus* **	**BCC22358**	**OL322050**	**OL322037**	**–**	**OL322067**	**This study**
** *Ascopolyporus griseoperitheciatus* **	**BCC25788**	**OL322051**	**OL322038**	**OL322058**	**OL322068**	**This study**
** *Ascopolyporus khaoyaiensis* **	**BCC43314**	**OL322052**	**OL322039**	**–**	**OL322069**	**This study**
** *Ascopolyporus khaoyaiensis* **	**BCC43741**	**OL322041**	**OL322040**	**–**	**OL322070**	**This study**
** *Ascopolyporus polychrous* **	**P.C. 546**	**DQ118737**	**DQ118745**	**DQ127236**	**–**	[15]
** *Ascopolyporus purpuratus* **	**BCC88388**	**OL322046**	**OL322033**	**–**	**OL322064**	**This study**
** *Ascopolyporus purpuratus* **	**BCC88389**	**OL322047**	**OL322034**	**–**	**–**	**This study**
** *Ascopolyporus purpuratus* **	**BCC88430**	**OL322045**	**OL322032**	**OL322059**	**OL322063**	**This study**
*Ascopolyporus villosus*	ARSEF 6355	AY886544	DQ118750	DQ127241	–	[14,15]
*Beauveria bassiana*	ARSEF 300	–	AY531924	HQ880831	HQ880903	[33,48]
*Beauveria kipukae*	ARSEF 7032	–	HQ881005	HQ880875	HQ880947	[48]
*Beauveria staphylinidicola*	ARSEF 5718	EF468836	EF468776	EF468881	–	[45]
*Beauveria varroae*	ARSEF 2694	–	HQ881004	HQ880874	HQ880946	[48]
*Blackwellomyces cardinalis*	OSC 93609	AY184962	DQ522325	DQ522370	DQ522422	[49,50]
*Blackwellomyces cardinalis*	OSC 93610	AY184963	EF469059	EF469088	EF469106	[45,50]
*Blackwellomyces pseudomilitaris*	BCC 1919	MF416534	MF416478	–	MF416440	[44]
*Blackwellomyces pseudomilitaris*	BCC 2091	MF416535	MF416479	–	MF416441	[44]
*Cordyceps bifusispora*	EFCC 5690	EF468806	EF468746	EF468854	EF468909	[45]
*Cordyceps lepidopterorum*	TBRC 7263	MF140699	MF140819	MF140768	MF140792	[51]
*Cordyceps piperis*	CBS 116719	AY466442	DQ118749	DQ127240	EU369083	[15,17]
*Cordyceps takaomontana*	BCC28612	FJ765252	FJ765268	–	–	[52]
*Cordyceps tenuipes*	ARSEF 5135	JF415980	JF416020	JN049896	JF416000	[53]
*Engyodontium aranearum*	CBS 309.85	AF339526	DQ522341	DQ522387	DQ522439	[49]
*Gibellula leiopus*	BCC 16025	MF416548	MF416492	MF416649	–	[44]
*Gibellula pulchra*	NHJ 10808	EU369035	EU369018	EU369056	EU369076	[17]
*Gibellula* sp.	NHJ 10788	EU369036	EU369019	EU369058	EU369078	[17]
*Gibellula* sp.	NHJ 13158	EU369037	EU369020	EU369057	EU369077	[17]
*Hevansia arachnophila*	NHJ 10469	EU369031	EU369008	EU369047	–	[17]
*Hevansia cinerea*	NHJ 3510	–	EU369009	EU369048	EU369070	[17]
*Hevansia nelumboides*	BCC 2093	MF416530	MF416473	–	MF416437	[44]
*Hevansia novoguineensis*	NHJ 4314	–	EU369012	EU369051	EU369071	[17]
*Hyperdermium caulium*	AF242354	AF242354	–	–	–	[11]
*Hyperdermium pulvinatum*	P.C. 602	DQ118738	DQ118746	DQ127237	–	[15]
*Lecanicillium antillanum*	CBS 350.85	AF339536	DQ522350	DQ522396	DQ522450	[45]
*Lecanicillium psalliotae*	CBS 363.86	AF339559	EF468784	EF468890	–	[45]
*Lecanicillium psalliotae*	CBS 532.81	AF339560	EF469067	EF469096	EF469112	[45]
*Neotorrubiella chinghridicola*	BCC 39684	MK632096	MK632148	MK632071	MK632181	[30]
*Neotorrubiella chinghridicola*	BCC 80733	MK632097	MK632149	MK632072	MK632176	[30]
*Samsoniella aurantia*	TBRC 7271	MF140728	MF140846	MF140791	–	[51]
*Samsoniella aurantia*	TBRC 7273	MF140726	MF140844	–	MF140816	[51]
*Samsoniella inthanonensis*	TBRC 7915	MF140725	MF140849	MF140790	MF140815	[51]
*Samsoniella inthanonensis*	TBRC 7916	MF140724	MF140848	MF140789	MF140814	[51]
**Outgroup**						
*Flavocillium bifurcatum*	YFCC 6101	MN576781	MN576951	MN576841	MN576897	[54]
*Lecanicillium* sp.	CBS 639.85	KM283801	KM283824	KM283843	KM283865	[54]

**Table 2 jof-08-00516-t002:** Morphological comparisons of *Ascopolyporus* and related species. NA, not applicable.

Name	Host	Stromata (mm)	Perithecia (µm)	Asci (µm)	Ascospores (µm)	Conidiogenous Cell (µm)	Conidia (µm)	References
*Ascopolyporus albus*	Scale insect,	pulvinate,	semi-immersed,	hyaline, cylindrical,	hyaline, filiform,	solitary,	enteroblastic,	This study
	Epiphyte	subglobose to globose	obpyriform,	up to 250 × 3–4	whole, multiseptate,	slightly curved,	fusiform to acerose,	
		white to pinkish white	250–320 × 100–120		95–135 × 1	cylindrical,	1–4 septate,	
		3–6				up to 120 × 1–2	8–30 × 2–3.5	
*A. caulium*	Scale insect,	crustose, subcircular	cylindrical,	cylindrical to	filiform,	phialidic,	enteroblastic,	Sullivan et al., 2000
	Epiphyte	yellow to orange,	200–250 × 65–80	slightly fusiform,	multiseptate,	sparse layer	cylindrical to fusiform,	
		5–100		100–160 × 8–9	extending		slightly truncate at end,	
					to the length		aseptate: 5–7 × 1–1.5	
					of ascus × 1 wide		1–5 septate: 15–30 × 1.5–3	
*A. galloides*	Scale insect,	pulvinate, hemispherical,	semi-immersed,	hyaline, cylindrical,	hyaline, filiform,	solitary,	enteroblastic,	This study
	Epiphyte	upper: white-yellowish white	obclavate,	129–175 × 3–6	whole, aseptate,	slightly curved,	fusiform to acerose,	
		lower: strong orange	170–340 × 60–110		131–216 × 0.5	cylindrical,	aseptate to 1–4 septate,	
		1–7				30–294 × 1–2	5–27 × 2–4	
*A. griseoperitheciatus*	Scale insect	pulvinate, irregular pulvinate	semi-immersed,	hyaline, cylindrical,	hyaline, filiform,	solitary,	enteroblastic,	This study
	Epiphyte	to subglobose, 3–7	obovoid,	150–193 × 4–5	whole, aseptate,	slightly curved,	fusiform to acerose,	
		upper: very pale violet to	150–320 × 80–140		extending to the	cylindrical,	aseptate to 1–2 septate,	
		light purplish gray			length of ascus	acremonium-like,	4–17 × 2–4	
		lower: vivid yellow to orange				43–265 × 1–2		
*A. khaoyaiensis*	Scale insect,	flattened to convex,	semi-immersed,	hyaline, cylindrical,	hyaline, filiform,	solitary,	enteroblastic,	This study
	Epiphyte	cylindrical to irregular shaped,	obclavate,	up to 215 × 3–4	whole, aseptate,	slightly curved,	fusiform to acerose,	
		very pale violet to dark purple	300–360 × 100–120		175–200 × 1	cylindrical,	aseptate to 1–3 septate,	
		3–25				acremonium-like,	7–22 × 1.5–3	
						up to 50 × 1–2		
*A. philodendrus*	Scale insect,	subglobose,	immersed,	cylindrical,	filiform,	simple,	enteroblastic,	White et al., 2003
	Epiphyte	upper: sterile, red-purple	obclavate,	90–140 × 3–5	length of ascus	phialidic,	subcylindrical,	
		lower: fertile, white to tan	200–300 × 40–80			30–60 × 1–3	aseptate to 1–4 septate,	
		12–25					7–25 × 3–4	
*A. polychrous*	Scale insect,	polypore-like,	underside of stroma,	hyaline,	hyaline, filiform	NA	enteroblastic, oval,	Möller, 1901;
	Epiphyte	bright-rusty red or	immersed,	cylindrical,	to spiroid, 300 × 1,		1-multiseptate,	White et al., 2003
		white to yellow	narrow obclavate,	500 × 4	disarticulate into		7–12 × 4–6	
		40	up to 750		part-spores, 6 × 1			
*A. purpuratus*	Scale insect,	flattened to convex,	semi-immersed, ovoid,	hyaline,	hyaline, filiform,	solitary,	enteroblastic,	This study
	Epiphyte	yellow to very pale,	300–420 × 100–150	cylindrical,	whole, aseptate,	slightly curved,	fusiform to acerose,	
		purple-violet,		up to 240 ×	100–220 × 1–	cylindrical,	aseptate,	
		5–12 × 3–8		2–4	1.5	acremonium-like,	5–18 × 1.5–3	
						up to 55 × 1–2		
*A. villosus*	Scale insect,	white to pale yellow,	not produced	NA	NA	phialidic	enteroblastic,	White et al., 2003; Bischoff et al., 2005
	Epiphyte	12–25					subcylindrical, guttulate,	
							aseptate to 1–4 septate,	
							10–22 × 2–5	
*Neohyperdermium piperis*	Scale insect,	pulvinate, subglobose	immersed, obpyriform	cylindrical,	filiform,	upright,	hyaline, subcylindrical,	Bischoff and White, 2004
	Epiphyte	to cylindrical,	to cymbiform,	120–170 × 3–5	disarticulating into	verticillate,	rarely subglobose,	
		white to yellow, 3–10 × 3–6	175–290 × 40–80		part-spores, 4–9 × 1–2	150–400	aseptate, 3–5 × 1–2	
*N. pulvinatum*	Scale insect,	pulvinate, white to tan,	cone-shaped,	linear,	filiform,	phialidic, hyaline,	enteroblastic, subcylindrical,	Sullivan et al., 2000
	Epiphyte	3–6	150–250 × 100–130	150–240 × 5–7	multiseptate,	1 to several septa,	slightly arcuate,	
					130–225 × 1	40–100 × 1–3	aseptate: 14–16 × 2.5–3	
							1–5 septate: 22–30 × 2.5–4	

## Data Availability

Publicly available datasets were analyzed in this study. These data can be found in: Genbank, https://www.ncbi.nlm.nih.gov/genbank/ (accessed on 9 November 2021; MycoBank, https://www.mycobank.org/ (accessed on 9 November 2021); Index Fungorum, http://www.indexfungorum.org/Names/Names.asp (accessed on 5 December 2021).

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
