# Peer review of "Is Hyperdermium Congeneric with Ascopolyporus? Phylogenetic Relationships of Ascopolyporus spp. (Cordycipitaceae, Hypocreales) and a New Genus Neohyperdermium on Scale Insects in Thailand"

_jof, 2022, doi:10.3390/jof8050516_

Round 1
Reviewer 1 Report
Dear authors and editors,
Here is the review of the manuscript entitled "Is Hyperdermium congeneric with Ascopolyporus? Phylogenetic relationships of Ascopolyporus spp. (Cordycipitaceae, Hypocreales) and a new genus Neohyperdermium on scale insects in Thailand." written by Natapol Pornputtapong & co-authors.
The paper aims to describe five new species of insect-pathogenic fungi from the genus Ascopolyporus found in Thailand. All species were analyzed by integrative taxonomic methods on the basis of their morphological and molecular characters (LSU, TEF1, RPB1 and RPB2 marker genes). Generic positions of newly described species were analyzed as well. In phylogenetic tree provided, the type species of Hyperdermium (H. caulium) clustered within the genus Ascopolyporus suggesting Hyperdermium is a synonym of Ascopolyporus. The new genus Neohyperdermium is proposed for other species originally assigned to Hyperdermium (distant from the type species of the genus) and Cordyceps occurring on scale insects and host plants as epiphytes, and thus two new combinations (N. pulvinatum and N. piperis) were proposed. Taxonomic identification keys for Neohyperdermium and Ascopolyporus were provided.
Authors followed the newest International code of nomenclature for algae, fungi, and plants. Morphological descriptions and molecular analysis used in the study are well done. Geographical coordinates of holotype collections should be added. Other than that the manuscript is of a very good quality.
All suggested minor corrections/additions are included in the attached review of the manuscript file (pdf).
Best, Reviewer

Author Response
Please see attachment. We have answered the comments and suggestions of Reviewer 1 in the pdf file and have incorporated these changes in the word file that will be attached to Reviewer 2.

Reviewer 2 Report
The authors have conducted interesting work on the phylogenetic relationship in Ascopolyporus using morphological and molecular evidences. However, this study is not yet ready to be published and it must pass through major revision that should improve this manuscript.
Below the authors will find some comments that will help to improve the article.
My biggest question regarding the study is the following: what is the relevance of analyzing the phylogenetic relationships in Ascopolyporus? It should be clear to readers.
The title is too long, please shorten it.
Abstract
Line 21: no space between “Ascopolyporus. Moreover….”
Line 22: no space H. caulium
Introduction
The introduction focuses a lot on unnecessary details, e.g.
line 35: why the authors mentioned: “…associated with aphids…………”?
line 37: what for the authors wrote in the bracket seven families? And for the record, they only mentioned about six families, not seven.
Lines 80-87: this paragraph should give information about why the new species is suspected to the undescribed.
Line 80: Ascopolyporus/Hyperdermium is not clear. Before using it the authors should explain why they used such a provision. It is not clear especially when the authors mention that they study Ascopolyporus species.
Lines 82-83: What made the authors so sure that these species were not members of Hyperdermium genus?
Line 84: “….of these two genera…” is not clear. Please, provide the name of genera.
Materials and Methods
Collection and Isolation
Please specify the number of specimens per species and location used in this study. How many species? How many locations? I suggest authors to prepare the Table with the list of investigated species with collection details (exact location, latitude and longitude of the species collection sites).
DNA extraction, PCR, and sequencing
Lines 119-120: Please describe what kind of modification of CTAB protocol was used for DNA extraction.
Lines 120-127: Please provide the names of primers for each marker and conditions of PCR amplification and sequencing in the text. Were the PCR products cleaned?
Lines 122-125: Dejonized water is missed in the reaction mix of PCR.
Line 127 – Please add the sentence including the information about your sequences deposited in the GenBank under Accession Numbers XXXXX- XXXXX.
Sequence alignment and phylogenetic analyses
Line 131: Please provide references for “previous studies”. How many sequences from GenBank for each marker were used? Please add this information to the text. Additionally, the authors should mention the details they showed in Table 1.
Lines 133-134: Why did the authors analyse only LSU as a single-locus by ML and BI? Did they construct trees for each marker and compare each other with the concatened sequences?
Line 137: The model evolution is also used in ML analyses. This information should be provided before the description of ML analyses.
Line 139: It is not clear why the authors used AliView to determine the model of evolution. Please provide a model of evolution for each marker separately.
Line 144: In BI analyses nodes are supported with posterior probability, not bootstrap values.
Results
Lines 147-149: This information is not a result. It should be moved into the Material & methods section.
Lines 149-151: For which datasets trees were congruent?
Lines 151-152: Please, provide the names of these five novel species. Why the authors describe only multigene phylogenetic analyses? What about LSU phylogenetic analyses?
Line 155: Clade can be divided into subclades. Please, correct it.
Lines 162-165: Why did the authors use only H. caulium? What about H. pulvinatum?
Table S1: I do not understand why the authors duplicate information from Table 1. Table S1 should be deleted.
Lines 165-169: I cannot see that Hyperdermium is closely related to Blackwellomyces in Fig. S1.
Lines 169-171: Please, clarify that you described the tree in Fig. 1. Why did not you use H. pulvinatum in Fig. S1?
Lines 171-172: It is not the results. The authors suggest to “transfer these two species to the genus Neohyperdermium.” I would recommend caution at this point because for sure would require more evidence.
The biggest problem is that the trees Fig. 1 and Fig S1 are not congruent. The phylogenetic relationships within Ascopolyporus are different when compared both trees. In Fig. S1 clade is not divided into two distinct subclades pulvinate and flattened from Fig. 1.
Table 1: I suggest to present species not alphabetically. The outgroup species should be at the end of the table. I would add one additional column “Family” or something like that. Hyperdermium caulium should be added to this table.
Figure 1: In the description of the tree it should be added what is in the pictures in the Ascopolyporus clade. I cannot see Cordycipitaceae on the tree. Please, clarify it.
Figure 2: It should explain what is it BBH30734, the abbreviation OA, PDA.
Lines 239-240: Please, explain what is it 91D, 158, 25A.
Discussion
Lines 555-558: Please provide specimens of which species. It is not a truth that the trees in Fig 1 and Fig. S1 are congruent.
Lines 561-562: It is not clear to me why the authors mentioned three different species of Hyperdemium.
Line 568: do not use the italics “and”.
Lines 570-572: In Table 2 the authors did not write that A. vilosus and A. polychrous possess completely immersed perithecia.
Line 594: Please explain what is IPF.
Line 601: should be “which”
Author Response
Here are our answers to Reviewer 2's comments and suggestions:
#Reviewer 2
The authors have conducted interesting work on the phylogenetic relationship in Ascopolyporus using morphological and molecular evidences. However, this study is not yet ready to be published and it must pass through major revision that should improve this manuscript.
Below the authors will find some comments that will help to improve the article.
My biggest question regarding the study is the following: what is the relevance of analyzing the phylogenetic relationships in Ascopolyporus? It should be clear to readers.
The title is too long, please shorten it.
We believe the title represents what we wanted to investigate, and Reviewer 1 did not comment on it so we would like to keep it as it is.
Abstract
Line 21: no space between “Ascopolyporus. Moreover….”
- Done as suggested, in line 21.
Line 22: no space H. caulium
- Done as suggested, in line 22.
Introduction
The introduction focuses a lot on unnecessary details, e.g.
line 35: why the authors mentioned: “…associated with aphids…………”?
- Aphids, whiteflies, and scale insects are classified in the same suborder Sternorrhyncha, order Hemiptera. Their taxonomy is based mainly on the microscopic cuticular features of the adult female.
line 37: what for the authors wrote in the bracket seven families? And for the record, they only mentioned about six families, not seven.
- Corrected as suggested, in line 39.
Lines 80-87: this paragraph should give information about why the new species is suspected to the undescribed.
- We added to the sentences “During our continuous survey of insect-pathogenic fungi (IPF) in national parks and community forests in Thailand, we encountered hyperdermium-like specimens with differences in phenotypic characters including colors and sizes of stromata. These morphologically diverse specimens were preliminarily identified as members of the genus Hyperdermium and Ascopolyporus. (Line 80-84)
Line 80: Ascopolyporus/Hyperdermium is not clear. Before using it the authors should explain why they used such a provision. It is not clear especially when the authors mention that they study Ascopolyporus species.
- We elaborated on the similarities of these 2 genera in the paragraph before this, on lines 55-62. The distinction between Ascopolyporus and Hyperdermium is morphologically not clear and one can easily misidentify one from the other. Additionally, other studies have already shown that members of Hyperdermium do not form a monophyletic clade. Hence, with the addition of more Ascopolyporus species and inclusion of the type species of both genera, our study here has clearly shown that even the type species of Hyperdermium is found in the Ascopolyporus
Lines 82-83: What made the authors so sure that these species were not members of Hyperdermium genus?
- We have confirmed our results by using molecular technique. All samples were clustered within the Ascopolyporus clade as you can see in new analysis of phylogenetic tree.
Line 84: “….of these two genera…” is not clear. Please, provide the name of genera.
- We mentioned the genera Hyperdermium and Ascopolyporus.
Materials and Methods
Collection and Isolation
Please specify the number of specimens per species and location used in this study. How many species? How many locations? I suggest authors to prepare the Table with the list of investigated species with collection details (exact location, latitude and longitude of the species collection sites).
- Done as suggested, in line 95-101. We have provided all specimens data of each species in “Additional specimen examined”
DNA extraction, PCR, and sequencing
Lines 119-120: Please describe what kind of modification of CTAB protocol was used for DNA extraction.
- CTAB protocol is usually used for extracting DNA from plants. We have modified CTAB protocol by replacing 2-mercaptoethanol to polyvinylpyrrolidone in CTAB buffer, increasing temperature in the incubation process from 60 to 65 oC, from Doyle and Doyle, 1987 and this simple and efficient procedure for genomic DNA extraction from fungi was described in Thanakitpipattana et al., 2020.
Lines 120-127: Please provide the names of primers for each marker and conditions of PCR amplification and sequencing in the text. Were the PCR products cleaned?
- Information about the PCR primers and conditions were added in lines 131-142. PCR products were purified before sequencing.
Lines 122-125: Dejonized water is missed in the reaction mix of PCR.
- Done as suggested, in line 134.
Line 127 – Please add the sentence including the information about your sequences deposited in the GenBank under Accession Numbers XXXXX- XXXXX.
- Done as suggested, in lines 147-149.
Sequence alignment and phylogenetic analyses
Line 131: Please provide references for “previous studies”. How many sequences from GenBank for each marker were used? Please add this information to the text. Additionally, the authors should mention the details they showed in Table 1.
- Done as suggested. Reference information was added as an extra column in Table 1.
Lines 133-134: Why did the authors analyse only LSU as a single-locus by ML and BI? Did they construct trees for each marker and compare each other with the concatened sequences?
- Only LSU rRNA sequence from type species of Hyperdermium; caulium is available in the international databases.
Line 137: The model evolution is also used in ML analyses. This information should be provided before the description of ML analyses.
- ML analyses were subjected under the GTRCAT substitution model with 1000 rapid bootstrap replicates (line 154-159).
Line 139: It is not clear why the authors used AliView to determine the model of evolution. Please provide a model of evolution for each marker separately.
- We used Aliview only to convert file from fasta to nexus format, which is subsequently used to determine the model of evolution using MrModeltest in PAUP program. We did not run a separate analysis for each gene but to a concatenated alignment of multiple loci so we only have one model of evolution as a result in MrModeltest.
Line 144: In BI analyses nodes are supported with posterior probability, not bootstrap values.
- Posterior probability was added in the sentence (line 162).
Results
Lines 147-149: This information is not a result. It should be moved into the Material & methods section.
- The length of the alignment from our taxon sampling is a result and we believe this should be in the results section, not Materials and Methods.
Lines 149-151: For which datasets trees were congruent?
- We used the same dataset of combined gene to construct phylogenetic tree by using ML and BI. Then compared each topology between both methods, which is congruent.
Lines 151-152: Please, provide the names of these five novel species. Why the authors describe only multigene phylogenetic analyses? What about LSU phylogenetic analyses?
- We have mentioned the name of these five novel species in lines 179-183.
Line 155: Clade can be divided into subclades. Please, correct it.
- Done as suggested.
Lines 162-165: Why did the authors use only H. caulium? What about H. pulvinatum?
- caulium is the type species of Hyperdermium but only has LSU sequence data available. Hence, we needed to run an analysis that included the type with only this data. H. pulvinatum has been shown to be distantly related to H. caulium in other studies (Sung et al., 2007; Kepler et al., 2017; Wang et al., 2020) (Line 608-614).
Table S1: I do not understand why the authors duplicate information from Table 1. Table S1 should be deleted.
- We removed the supplementary data as we have revised the phylogenetic tree in Fig 1 to include H. caulium already to show its position within Ascopolyporus
Lines 165-169: I cannot see that Hyperdermium is closely related to Blackwellomyces in Fig. S1.
- This is clearly shown now in Fig. 1 using multi-gene phylogeny. We removed all supplementary materials
Lines 169-171: Please, clarify that you described the tree in Fig. 1. Why did not you use H. pulvinatum in Fig. S1?
- The original code “P.C. 602” in Fig. 1 is pulvinatum and we removed all supplementary materials
Lines 171-172: It is not the results. The authors suggest to “transfer these two species to the genus Neohyperdermium.” I would recommend caution at this point because for sure would require more evidence.
- Our phylogenetic study based on multiple genetic loci has shown that 2 species, H. pulvinatum and C. piperis, do not belong to Ascopolyporus. Since we have shown that the type of Hyperdermium, caulium, is nested within the Ascopolyporus clade, the genus is conspecific to Ascopolyporus. This subsequently facilitates the transfer of H. pulvinatum and C. piperis to another genus.
In this point, the molecular evidences
The biggest problem is that the trees Fig. 1 and Fig S1 are not congruent. The phylogenetic relationships within Ascopolyporus are different when compared both trees. In Fig. S1 clade is not divided into two distinct subclades pulvinate and flattened from Fig. 1.
- Fig 1 was based on multiple loci while Fig. S1 was based only on LSU sequence. We revised Fig. 1 by adding caulium, the type species of Hyperdermium to the multi-gene analysis even if this only contains the LSU sequence. Despite the missing data in the analysis it still shows that H. caulium is nested within the Ascopolyporus clade, which is what we wanted to show in Fig S1. The reason why we initially added a supplementary Fig based on LSU data to show the taxonomic position of H. caulium and that it belongs to the Ascopolyporus clade. However, we now removed the supplementary material to avoid confusion.
Table 1: I suggest to present species not alphabetically. The outgroup species should be at the end of the table. I would add one additional column “Family” or something like that. Hyperdermium caulium should be added to this table.
- The outgroup is placed at the end of the Table 1 as suggested. All species or strains belong only to the Cordycipitaceae so we believe that there is no need to add an additional column ‘Family’
Figure 1: In the description of the tree it should be added what is in the pictures in the Ascopolyporus clade. I cannot see Cordycipitaceae on the tree. Please, clarify it.
- Fig 1 represents members of the Cordycipitaceae where Ascopolyporus also belongs. This information is added in the Figure legend.
Figure 2: It should explain what is it BBH30734, the abbreviation OA, PDA.
- BBH30734 is holotype specimens of albus. We have mentioned about the holotype of each species in part of typification, and abbreviation of BBH is mentioned in collection and isolation (2.1) (Line 106).
- The abbreviation of OA for oatmeal agar and PDA for potato dextrose agar were mentioned in Materials and methods section, including their definition and recipe; morphology study (Lines 110-112).
Lines 239-240: Please, explain what is it 91D, 158, 25A.
- We have used the standard color chart for color comparisons (RHS colour chart), which is given the closely color shade with our specimens, also mentioned in the Methods section.
Discussion
Lines 555-558: Please provide specimens of which species. It is not a truth that the trees in Fig 1 and Fig. S1 are congruent.
- We replaced a Fig 1 that already includes caulium and shows its position in the Ascopolyporus clade.
Lines 561-562: It is not clear to me why the authors mentioned three different species of Hyperdemium.
- There are 3 different species of Hyperdermium in literature. Hyperdermium caulium is the type species while other known species are pulvinatum and H. bertonii. However, H. bertonii was proven to be the conspefic (same species as) with H. caulium (Chaverri et al., 2008)
Line 568: do not use the italics “and”.
- Done as suggested.
Lines 570-572: In Table 2 the authors did not write that A. vilosus and A. polychrous possess completely immersed perithecia.
- In the discussion we mentioned that philodendrus and A. polychrous possess completely immersed perithecia while A. vilosus was reported not to produce any perithecia on stromata
Line 594: Please explain what is IPF.
- We added the full name of IPF to insect-pathogenic fungi in Line 615.
Line 601: should be “which”
- Done as suggested.

Round 2
Reviewer 2 Report
The MS can be accepted after minor revision. Authors can find my comments in the attached manuscript file.

Author Response
Dear Reviewer2,
First of all, we are grateful for your comments and suggestions that allowed us to greatly improve the quality of the manuscript.
We agree with all your comments, and we corrected point by point the manuscript accordingly. The changes based on these comments and suggestions were marked with track changes in word file. Please see the attachment.
Sincerely,
Jennifer Luangsa-ard